# DEEPVERSE: 4D AUTOREGRESSIVE VIDEO GENERATION AS A WORLD MODEL

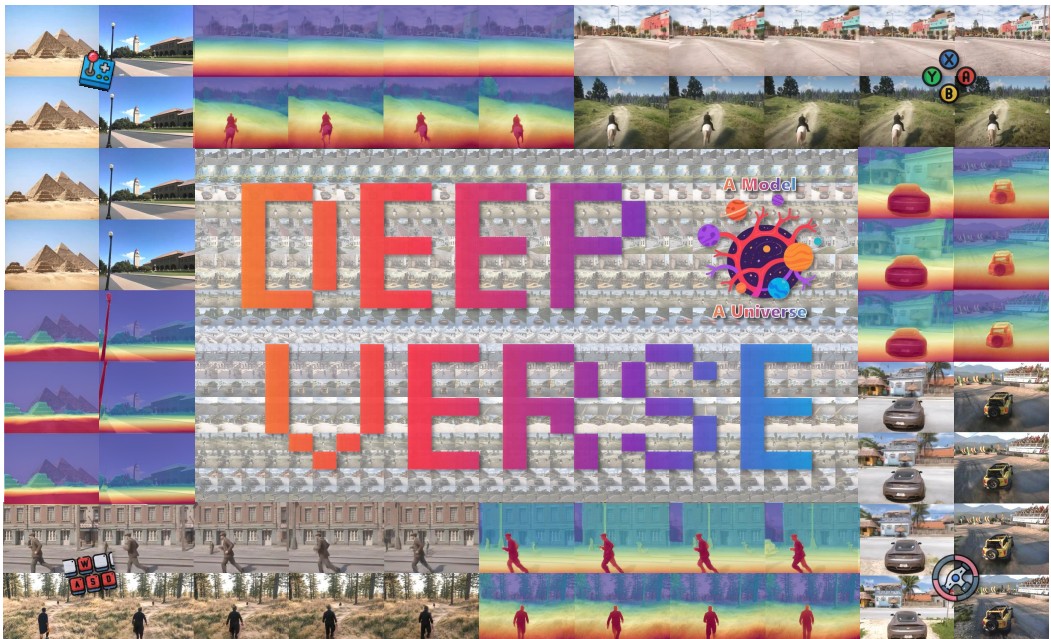

Figure 1: We introduce DeepVerse, an interactive world model grounded in 4D autoregressive video generation. By establishing a 4D spatiotemporal distribution of the world, DeepVerse enables continuous and coherent 4D future prediction from merely a single input image, effectively modeling both spatial layouts and temporal dynamics simultaneously.

## ABSTRACT

World models serve as essential building blocks toward Artificial General Intelligence (AGI), enabling intelligent agents to predict future states and plan actions by simulating complex physical interactions. However, existing interactive models primarily predict visual observations, thereby neglecting crucial hidden states like geometric structures and spatial coherence. This leads to rapid error accumulation and temporal inconsistency. To address these limitations, we introduce DeepVerse, a novel 4D interactive world model explicitly incorporating geometric predictions from previous timesteps into current predictions conditioned on actions. Experiments demonstrate that by incorporating explicit geometric constraints, DeepVerse captures richer spatio-temporal relationships and underlying physical dynamics. This capability significantly reduces drift and enhances temporal consistency, enabling the model to reliably generate extended future sequences and achieve substantial improvements in prediction accuracy, visual realism, and scene rationality. Furthermore, our method provides an effective solution for geometry-aware memory retrieval, effectively preserving long-term spatial consistency. We validate the effectiveness of DeepVerse across diverse scenarios, establishing its capacity for high-fidelity, long-horizon predictions grounded in geometry-aware dynamics.

## 1 INTRODUCTION

Interactive understanding of the physical world is a fundamental task for intelligent systems. World models, which aim to learn state transition functions from raw observations of external environments, provide essential predictive capabilities for intelligent agents, enabling them to imagine future states, evaluate possible actions, and navigate complex, dynamic scenarios. Recent progress in world models has demonstrated considerable potential in tasks such as visual simulation (NVIDIA et al., 2025), embodied navigation (Team et al., 2025), and manipulation (Zhen et al., 2025).

Despite notable advancements in constructing effective world models, current online approaches (Feng et al., 2024; Valevski et al., 2024; Decart et al., 2024) still suffer significantly from cumulative prediction errors and the forgetting issue. Addressing the above challenges is non-trivial. Most existing methods (Song et al., 2025a; Xiao et al., 2025) attempt to mitigate these issues by developing sophisticated techniques to efficiently incorporate historical frames. For instance, FramePack (Zhang & Agrawala, 2025) compresses past frames into a fixed-length representation, thereby maintaining context within a transformer's limited memory. However, these visual-centric strategies fundamentally overlook a critical aspect: videos inherently represent 2D projections of a dynamic 3D/4D physical world. Without explicit modeling of underlying geometric structures, models inevitably struggle to maintain long-term accuracy and consistency in visual predictions.

To address this core challenge, we introduce DeepVerse, a novel autoregressive world model that directly learns the dynamics of the underlying 4D world. Unlike methods that predict future pixels, DeepVerse operates on 4D states, grounding its predictions in a consistent geometric reality. This approach mitigates key issues in purely visual autoregressive systems, such as scale ambiguity (Fig. 3a), and directly addresses the problems of drifting and forgetting prevalent in conventional methods. Specifically, shifting the predictive target from pixels to 4D states inherently mitigates error accumulation (drift) by enforcing geometric constraints on spatial correlations. Furthermore, to combat long-term forgetting, we propose a global 4D memory bank that stores all historical states aligned to a common initial coordinate frame. During inference, DeepVerse retrieves relevant past states from this memory based on similarity to the current viewpoint. This retrieved context conditions the next prediction, ensuring the model maintains spatiotemporal coherence over extended sequences without catastrophic forgetting.

To sum up, our contributions can be summarized as follows:

- We introduce DeepVerse, an autoregressive framework for 4D world modeling, establishing foundational guidelines for architectural design in future interactive world model development.
- We innovatively incorporate 4D information into an autoregressive world modeling framework. By modeling the 4D world explicitly during generation, our approach significantly improves visual consistency and resolves scale ambiguity issues common in unimodal visual paradigms.
- Building upon the DeepVerse framework's capability for concurrent spatial distribution modeling, we have engineered a spatial memory mechanism to enhance long-term temporal consistency in generated sequences, ensuring spatiotemporal continuity in autoregressive generation.

## 2 METHOD

To provide a comprehensive illustration of our method, we elaborate on our problem formulation in section 2.1 and discribe the methodology for tailoring model architecture in section 2.2. The construction of training datasets is systematically addressed in Section 2.3. Finally, Section 2.4 demonstrates DeepVerse's operational workflow during the inference phase.

### 2.1 PROBLEM FORMULATION

World models aim to learn the transition function $P(s_{t+1}|a_t, s_t)$ in a Markov Decision Process (MDP), where $s_t$ denotes the environment state at timestamp $t$. However, many real-world applications are fundamentally Partially Observable MDPs (POMDPs), where the true state $s_t$ is latent. Prior works (Bruce et al., 2024; Valevski et al., 2024; Decart et al., 2024) often rely directly on visual observations $v_t$ as input. However, this signal is incomplete and non-Markovian, which can degrade model performance. To address this limitation, DeepVerse introduces a composite 4D state

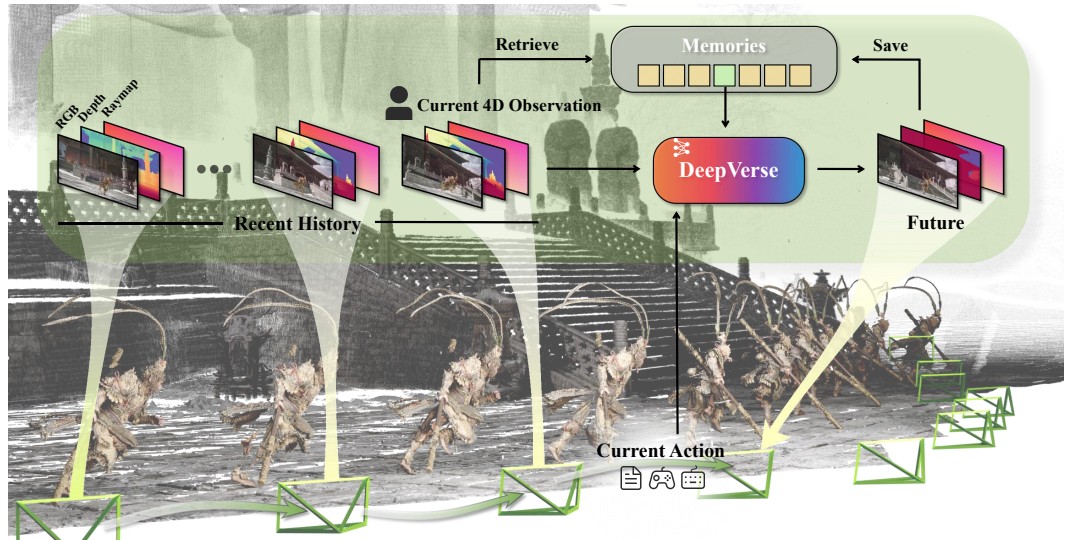

Figure 2: **Our framework.** The inputs to DeepVerse consist of: (1) a sequence of $m$ consecutive 4D observations encompassing current and recent estimated states; (2) spatial conditions retrieved from a global memory pool through the selective mechanism $\psi$; (3) textually specified control signals. The system subsequently generates $k$ temporally coherent 4D future observations, which are automatically archived into the global memory repository for persistent world state tracking.

representation, $\hat{s}_t$, as a more effective proxy for the latent state $s_t$:

$$\hat{s}_t = (v_t, g_t). \tag{1}$$

Here, $v_t$ is the visual observation, while $g_t$ encapsulates geometric information, specifically the camera $c_t$ and depth map $d_t$. By fusing visual and geometric data, each $\hat{s}_t$ forms a local 3D representation of the scene. To model the dynamics of this representation and mitigate the long-term drift caused by non-Markovian observations in POMDPs, DeepVerse employs an adaptive memory architecture. The task is framed as an auto-regressive sequential prediction problem:

$$f_\theta = P\left(\hat{s}_{t+1:t+k} \mid a_t, \underbrace{\hat{s}_t, \hat{s}_{t-m:t-1}}_{\text{Recent Context}}, \underbrace{\psi\left(\hat{s}_{0:t-m-1}\right)}_{\text{Long-term Memory}}\right). \tag{2}$$

As shown in Eq. 2, our predictive model $f_\theta$ conditions its output on three components: (1) the current action $a_t$ and state representation $\hat{s}_t$; (2) a **recent context** of the last $m$ representations, $\hat{s}_{t-m:t-1}$, to capture short-term dynamics; and (3) a **long-term memory** component to account for long-range dependencies. The function $\psi(\cdot)$ is a selective retrieval mechanism that fetches past representation from the historical buffer $\hat{s}_{0:t-m-1}$. This retrieved representation is the most relevant to the current state $\hat{s}_t$, helping the model to resolve ambiguities.

## 2.2 MODEL COMPONENTS

**4D Representation.** As formulated in Eq. 1, DeepVerse employs a 4D representation for state estimation. Specifically, each $g$ is a tensor with dimensions matching the input image, where each pixel stores a 3D coordinate (Wang et al., 2024b). We decompose 3D coordinates into depth $d_t$ and viewpoint components, which allows depth information to be directly encoded by the pre-trained Variational Autoencoder (VAE) (Kingma & Welling, 2022; Ke et al., 2024). Moreover, we parameterize depth values $d_t$ as the square root of disparity $e_t = \sqrt{1/d_t}$ (Song et al., 2025b). Finally, we adopt the raymap representation (Team et al., 2025; Chen et al., 2024b) to parameterize viewpoint $c$, which geometrically encodes camera orientation and position through ray direction vectors in the scene coordinate system. We construct $\hat{s}$ by channel-wise concatenating the three modalities. This unified structure ensures compatibility with standard image latents, enabling autoregressive future prediction through iterative generation.

**General Control.** Unlike previous works (Feng et al., 2024; Valevski et al., 2024) that integrated controller data as an additional training modality, our DeepVerse framework deliberately departs from

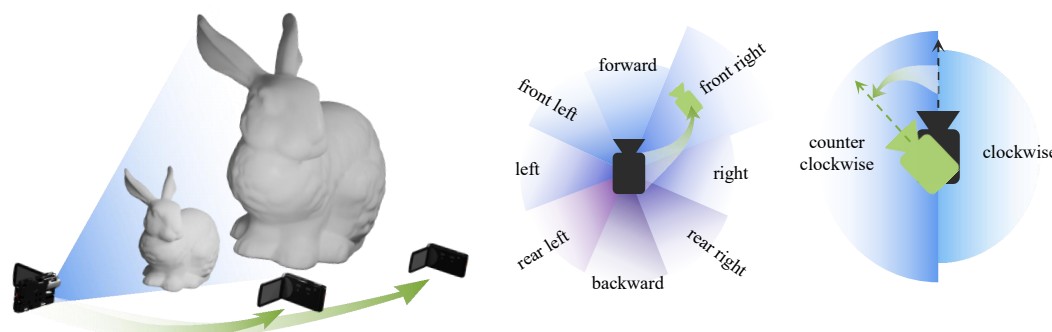

(a) Scale Ambiguity  (b) Perspective Movement  (c) Perspective Rotation

Figure 3: (a) Inferring 3D environments from a single image results in inherent scale ambiguity, a latent variable conditioned on the training data. Generating novel views from images alone, without 3D priors, is significantly more challenging than with explicit 3D structures, often leading to geometrically inconsistent extrapolations and error propagation in autoregressive predictions. (b)(c) Text descriptions of perspective changes can be algorithmically derived from camera pose variations.

this practice by relying solely on textual conditions. This strategy is motivated by two objectives: (1) maximizing the preservation of the pre-trained model's capabilities, and (2) leveraging the inherent versatility of textual control. This approach offers significant practical benefits: it allows for direct text-to-controller mapping in downstream applications and enables efficient fine-tuning for novel controllers.

**Spatial Condition.** Since we explicitly model the 4D representation, we maintain a memory pool of all past states $\{\hat{s}_0, \hat{s}_1, ..., \hat{s}_{t-1}\}$ to leverage historical context. The 3D coordinates of these states are aligned to the coordinate system of the initial frame $\hat{s}_0$, creating a unified global reference. For the current state $\hat{s}_t$, our spatial selection mechanism $\psi$ retrieves the most relevant historical state based on geometric proximity. This selection process, consists of two steps:

1. Candidate Selection by Proximity: First, we identify the $k$ historical states whose camera positions $T_i$ are closest to the current camera position $T_t$. This forms a candidate set $S$:

$$S = \underset{i \in \{0,...,t-1\}}{\overset{(k)}{\arg\min}} \|T_t - T_i\|^2 \tag{3}$$

2. Final Selection by Viewing Direction: From the candidate set $S$, we select the state $\hat{s}_j$ whose camera rotation $R_j$ is most aligned with the current rotation $R_t$, measured by minimizing the angle between rotation matrices:

$$j = \underset{i \in S}{\arg\min} \angle(R_t, R_i) \tag{4}$$

The selected state $\hat{s}_j$ is then encoded into a token sequence that serves as the spatial condition for processing the current state.

## 2.3 DATASET CONSTRUCTION

Our data collection process began with approximately 10 million frames of gameplay footage, from which we systematically removed all UI elements using ReShade (Reshade, 2024). Subsequently, leveraging an existing automatic camera annotation pipeline (Team et al., 2025), we synthesized datasets containing precise intrinsic and extrinsic camera parameters, depth maps, and high-fidelity synthetic images. The acquired camera parameters were first used to filter out potentially erroneous or low-quality data. To facilitate interactive applications, we implemented a hierarchical annotation protocol, applying textual labels at the video clip level and more detailed, motion-specific labels to finer-grained segments of frames.

**Filtering Criteria.** Excessive camera rotation or rapid view transitions degrade reconstruction quality after 3D VAE-based encoding-decoding. To mitigate this, we introduce a chunk-wise filtering criterion: the chunk size equals the VAE's temporal compression ratio. A video clip is valid only if the cumulative rotation angle across chunks remains below a threshold $\delta_{rot}$. This angle is computed

as the angular difference in the forward direction between the last frames of consecutive chunks. We also exclude clips with insufficient camera or character movement by deriving displacement from camera extrinsic parameters, discarding those below a threshold $\delta_{move}$.

**Caption Annotation.** Given the precise positional data obtained, we initially construct textual descriptions directly from camera movements/rotations as illustrated in Figure 3. Furthermore, we employ Qwen-VL (Bai et al., 2023) to annotate video clips, generating first-person descriptions for viewpoint transitions in egocentric videos, while creating third-person narratives for character actions and movements in exocentric recordings. We employ CLIP (Radford et al., 2021) and T5 (Raffel et al., 2023) to generate caption embeddings, adopting a methodology consistent with that implemented in SD3 (Esser et al., 2024).

**Training Preprocess.** For a video clip, global scene scaling based on scene dimensions is implemented to ensure effective compression. To guarantee that the disparity values $d_i$ can be appropriately scaled into a constrained space for successful VAE encoding while preserving autoregressive causality, the entire sequence of $d_i$ is normalized by $d_{\max} \times \lambda$, where $\lambda$ serves as a modulation factor. This normalization ensures that the $d$'s range of the initial frame is transformed into $(0, \lambda] \subseteq [0, 1]$, thereby reserving value space for subsequent frames where $d_i$ may exceed $d_{\max}$. This mechanism effectively prevents truncation artifacts during the rescaling process to the VAE's input domain.

## 2.4 LONG-DURATION INFERENCE

To enable long-duration reasoning, we employ a sliding window approach (Song et al., 2025a; Chen et al., 2024a). Specifically, after generating the sequence $\hat{s}_{t:t+k}$, the last $m$ frames ($\hat{s}_{t+k-m+1:t+k}$) serve as the conditioning context for the next window. Prior to this process, a scaling operation is applied to the transitional segment: Using $\hat{s}_{t+k-m+1}$ as the initial frame of the next window, we compute $d_{\max}$ from $d_{t+k-m+1}$ to scale the parameters $d$ and $c$ within $\hat{s}_{t+k-m+2:t+k}$. The $d_{\max}$ value is saved to later rescale the output of the next window, guaranteeing that the final concatenated sequence maintains global consistency. The complete algorithm is detailed in 1.

---

**Algorithm 1:** Long-Duration Inference

**Input** : Observation $v_0$
**Output** : State sequence $\{\hat{s}_t\}_{t=1}^{\infty}$
Initialize memory $\mathbb{M} \leftarrow \{\hat{s_0} = (v_0, \mathbf{0}, \mathbf{0})\}$ ;
Initialize cache $\mathbb{C} \leftarrow \emptyset$ ;
**for** *Inference loops $i = 1, 2, ...$* **do**
    Read recent memories $\mathbb{C} \leftarrow Recent(\mathbb{M})$ ;
    Scale cache $\mathbb{C}$ ;
    **while** $Size(\mathbb{C}) < CacheMaxSize$ **do**
        Retrieve $s_{sp\hat{a}tial} \leftarrow \psi(\hat{s_{now}}||\mathbb{M})$ ;
        Read action $a_{now}$ ;
        $\hat{s}_{next} = f_\theta(\hat{a}_{now}, \mathbb{C}, \hat{s}_{spatial})$ ;
        Cache state $\mathbb{C} = \mathbb{C} \cup \{\hat{s}_{next}\}$
    Rescale cache $\mathbb{C}$ ;
    Update memories $\mathbb{M} = \mathbb{M} \cup \mathbb{C}$

---

## 3 EXPERIMENTS

DeepVerse is an autoregressive diffusion model based on flow matching (Liu et al., 2022; Albergo & Vanden-Eijnden, 2022; Lipman et al., 2022). To validate our design, we conducted comprehensive ablation studies. Specifically, as detailed in Section 3.1, we systematically investigated two distinct MM-DiT-based (Esser et al., 2024) architectures for historical information integration. Furthermore, to substantiate the necessity of 4D modality introduction in DeepVerse, a comparative analysis was conducted in Section 3.2, demonstrating the critical advantages of incorporating this novel modality. Finally, Section 3.3 presents the inference results of DeepVerse and its performance improvement over the baseline model.

## 3.1 DIFFERENT MODEL ARCHITECTURES

In the training paradigm of diffusion models, historical and future observations are first encoded into latent representations, which are subsequently patchified into tokens and input into a transformer-based (Vaswani et al., 2023) network. As illustrated in figure 4 (a), DeepVerse explores two approaches for injecting historical information based on the MM-DiT Esser et al. (2024) architecture. We first adopt GameNGen's (Valevski et al., 2024) methodology by directly concatenating historical information through channel-wise concatenation. Subsequently, inspired by existing video generation methods (Jin et al., 2024), we develop a token-wise concatenation strategy to integrate temporal information.

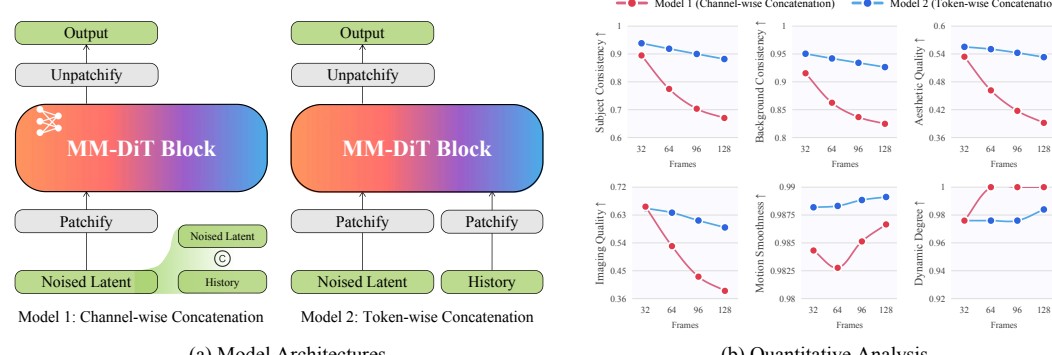

(a) Model Architectures       (b) Quantitative Analysis

Figure 4: (a) Two MM-DiT-based architectures were designed to inject historical information. (b) Quantitative evaluation results on VBench (Huang et al., 2024) demonstrate that Model 2 (Token-wise Concatenation) achieves better performance in nearly all metrics, exhibiting enhanced visual quality and reduced temporal drift issues compared to alternative architectures.

**Model 1: Channel-wise Concatenation.** First, individual frames in a video clip are encoded into latent representations, preserving the original temporal length. The latent states from different timesteps are then concatenated along the channel dimension. This combined representation is subsequently patchified into tokens for processing through the model. Finally, these tokens are unpatchified and decoded into outputs with target channel dimensions. A key advantage of this design is that it avoids creating additional tokens for conditioning frames. By integrating temporal information through channel concatenation (fusing noise tokens with historical states), it significantly reduces the computational overhead, particularly the non-linear FLOPs associated with the transformer's attention mechanism.

**Model 2: Token-wise Concatenation.** In contrast to Model 1, this architecture demonstrates distinct characteristics in latent processing: After video encoding into latent representations, temporal states from different timesteps are patchified independently to generate different tokens. This approach substantially increases token quantity, necessitating the implementation of a 3D VAE framework to achieve temporal compression in the latent space. The resulting latent representations exhibit reduced dimensionality along the temporal axis compared to the original video clip's frame count, maintaining a fixed temporal compression rate (except for the first frame) to balance information preservation and computational efficiency.

**Implementation Details.** Both comparative models maintain identical parameter scales of 2 billion. To enhance training efficiency, we implemented Fully Sharded Data Parallelism (FSDP) (Zhao et al., 2023) with ZeRO-2 optimization for both architectures. For Model 1, parameter initialization was performed using pre-trained weights from the SD3-medium (Esser et al., 2024), whereas Model 2 utilized Pyramid-Flow (Jin et al., 2024) initialization. Our experimental analysis revealed that excessive concatenation of historical frames in Model 1 failed to yield performance improvements, prompting adoption of a configuration with 7 historical frames and 1 noise frame. Moreover, we implement condition augmentation techniques (Ho et al., 2021) on the historical frames. Model 2 adheres to the Pyramid-Flow architecture's 57 frame protocol while implementing its dedicated 3D VAE for eightfold temporal compression. All training videos were standardized by first resizing them to $384p$ via bicubic interpolation and then center-cropping to a uniform $4:3$ aspect ratio. The corresponding camera intrinsic parameters in the metadata were adjusted accordingly. Notably, Model 1 demonstrates reduced token counts per computational step compared to Model 2, enabling deployment of larger global batch sizes - specifically 512 for Model 1 versus 256 for Model 2. Both architectures employed the AdamW optimizer (Loshchilov & Hutter, 2019) with cosine annealing learning rate scheduling, incorporating linear warm up during the initial $1\%$ of training iterations.

**Quantitative Results** The evaluation conducted on VBench (Huang et al., 2024) encompassed six metrics: *subject consistency*, *background consistency*, *aesthetic quality*, *imaging quality*, *motion smoothness*, and *dynamic degree*. Quantitative assessments were performed on the 32, 64, 96, and 128 generated frames, with comparative results graphically presented in figure 4b. Our findings reveal that Model 2's token-wise concatenation mechanism, despite introducing higher computational complexity (quantified as average GFLOPs of 1280.9 versus 1049.4 for Model 1), effectively mitigates

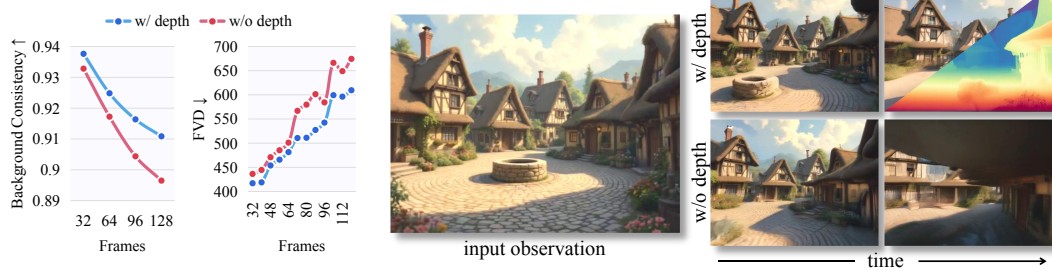

(a) Quantitative Analysis            (b) Visualization Analysis

Figure 5: **Ablation studies on depth modality.** (a) Quantitative results demonstrate that the integration of the depth modality yields better performance in FVD and consistency. (b) Qualitatively, models incorporating depth exhibit enhanced environmental comprehension, achieving improved visual quality and mitigating temporal drift artifacts compared to the baseline.

autoregressive model drift while achieving better visual performance. Notably, while channel-wise concatenation demonstrated competitive performance (Valevski et al., 2024; Alonso et al., 2024) in specific applications such as the DOOM gaming environment, our analysis suggests that temporal feature aggregation within single tokens exacerbates error accumulation phenomena, particularly under extended scenarios. This empirical evidence substantiates our architectural preference for token-wise concatenation, which demonstrates enhanced robustness across temporal dimensions in large-scale multimodal domains.

## 3.2 ABLATIONS

As elaborated in Section 3.1, we have innovatively introduced a novel modality into the DeepVerse framework. In this section, we train an additional model that aligns with conventional autoregressive methodologies by excluding the depth modality, retaining solely the raymap-based camera representation. Notably, the experimental configuration maintains identical training methodologies, datasets, and initialization parameters for corresponding layers across all compared models. For quantitative evaluation, we adopt the FVD Unterthiner et al. (2019) and VBench Huang et al. (2024) as principal assessment criteria.

**Introduction of New Modality.** We present a comparative visualization of two models in the figure 5, where both models generate predicted state sequences for future timesteps when initialized with a starting image and subjected to randomized action sequences. The empirical results demonstrate that the incorporation of depth modality substantially enhances the model's capacity to achieve comprehensive scene understanding, thereby enabling more precise estimation of latent world states that underlie observational inputs. This improved state estimation directly corresponds to enhanced visual predictive capabilities, as evidenced by our quantitative evaluation of synthesized video quality. While temporal drift persists as a fundamental challenge in autoregressive generation, our findings reveal that depth integration effectively alleviates this deterioration phenomenon, with measurable improvements observed both in quantitative metrics and visual representations.

**Spatial Memory.** By simultaneously predicting 3D camera poses during the generative process, we establish and maintain a global coordinate system anchored at the origin point defined by the initial frame's position. Our methodology implements a retrieval mechanism that queries the most recent pose from historical states to serve as spatial conditioning. During training, we strategically incorporate this spatial condition at controlled intervals as an additional modal constraint alongside textual inputs. For inference procedures, we adapt the InstructPix2Pix (Brooks et al., 2023) framework

Table 1: **Quantitative ablation study on depth modality.** Experiments conducted on six VBench metrics demonstrate that integrating depth modality achieves better results, confirming the critical influence of 3D information on visual quality within autoregressive video generation frameworks.

| | frames | subject consistency | background consistency | aesthetic quality | imaging quality | motion smoothness | dynamic degree |
|---|---|---|---|---|---|---|---|
| w/ depth (Ours) | 60 | **0.86939** | **0.92617** | **0.53415** | **0.48844** | **0.99032** | 1.00000 |
| | 120 | **0.81652** | **0.91087** | **0.50028** | **0.44639** | **0.99147** | 1.00000 |
| w/o depth | 60 | 0.83602 | 0.91899 | 0.49106 | 0.43774 | 0.98975 | 1.00000 |
| | 120 | 0.76812 | 0.89650 | 0.44095 | 0.37975 | 0.99062 | 1.00000 |

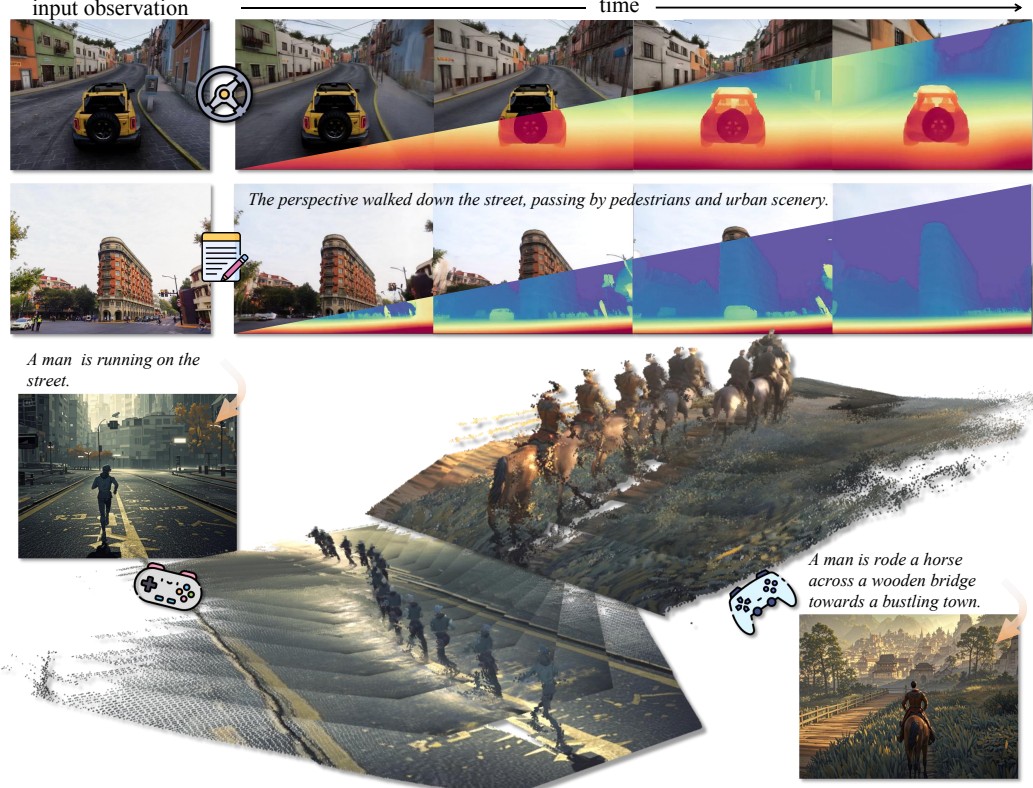

Figure 6: Visualization Results.

for conditional generation. As demonstrated in the figure 7, the integration of spatial conditioning enables extended temporal coherence in sequence generation that transcends the inherent limitations of fixed-duration video chunks, thereby achieving long-term spatial memory retention. To quantitatively validate this, we conducted experiments using identical action sequences, random seeds and initial frames, to compare the FVD of videos generated with and without spatial conditioning (cyclic trajectories are not enforced). The results, summarized in the table 7, demonstrate that spatial conditioning yields improved FVD, with more significant gains on longer inference sequences.

## 3.3 SIMULATION QUALITY

As illustrated in Figure 6, we demonstrate the capabilities of DeepVerse through comprehensive evaluations. For each experimental instance, we exclusively employ visual inputs as initial observations, which include game images, real-world images, and AI-generated images. Benefiting from our model's versatile conditioning mechanism, human-guided manipulations from various controllers can be manually projected into textual conditions for model input, while DeepVerse inherently supports direct textual condition integration. Our future prediction framework achieves highly consistent 4D representations while maintaining exceptional visual fidelity, with strict adherence to input conditions. Notably, the DeepVerse world model – grounded in 4D autoregressive video generation – distinguishes itself from conventional reconstruction-then-rerendering paradigms by simultaneously preserving viewpoint-object dynamics and predicting environmental interactions.

Furthermore, we compared the capabilities of DeepVerse and the base model (Jin et al., 2024) using VBench (Huang et al., 2024). Specifically, we selected a set of initial images from the test set and utilized QwenVL (Bai et al., 2023) to generate text prompts as conditions. As shown in the table 2, it can be observed that by modeling the 4D distribution, DeepVerse achieves better visual results.

Table 2: Quantitative comparison with PyramidFlow.

| | average score | subject consistency | background consistency | aesthetic quality | imaging quality | motion smoothness | dynamic degree |
|---|---|---|---|---|---|---|---|
| DeepVerse (Ours) | **0.848** | **0.899** | **0.946** | **0.578** | 0.726 | 0.990 | **1.000** |
| PyramidFlow | 0.762 | 0.897 | 0.944 | 0.552 | **0.733** | **0.994** | 0.500 |

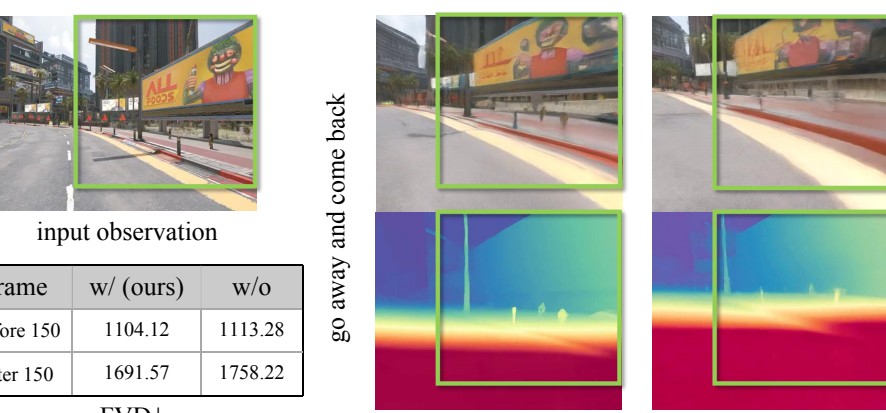

| frame | w/ (ours) | w/o |
|---|---|---|
| before 150 | 1104.12 | 1113.28 |
| after 150 | 1691.57 | 1758.22 |

FVD↓

w/ spatial condition (ours)   w/o spatial condition

Figure 7: The visualization of the spatial condition's effect.

## 4 RELATED WORKS

**Neural World Simulation.** Neural world simulation employs generative models for dynamic, interactive environments, simulating real-world physics, a common limitation in standard video generation. UniSim Yang et al. (2024b) tackles this by training an action-conditioned video model with multi-dimensional datasets, creating an interactive universal simulator. UniPi Du et al. (2023) reframes sequential decision-making as text-conditioned video generation, extracting control policies from generated future frames for cross-environment generalizability. Aether Team et al. (2025) argues videos are 2D projections and incorporates 3D structural information to better represent the underlying physical reality. Cosmos NVIDIA et al. (2025) shows that pre-training on physically-grounded video datasets, followed by fine-tuning, significantly enhances performance on physics-oriented AI tasks.

**Interactive Video Generation.** Interactive video generation merges interactivity with high-fidelity synthesis using neural networks. Methods like GameNGen Valevski et al. (2024), Oasis Decart et al. (2024), DIAMOND Alonso et al. (2024), and GameFactory Yu et al. (2025) achieve controllability by incorporating control labels into training datasets. Genie Bruce et al. (2024) introduces a Latent Action Model (LAM) to learn generalized actions from videos. GameGen-X Che et al. (2024) first pretrains on text-video pairs and then adapts to other control modalities. WorldMem Xiao et al. (2025) leverages 3D pose representations for memory retrieval to improve long-term consistency.

**3D/4D Representations.** The integration of 3D and 4D representations Mildenhall et al. (2021); Zhu (2023); Kerbl et al. (2023); Wu et al. (2024); Zhu et al. (2023); Yang et al. (2024a); Wang et al. (2024b); Zhang et al. (2024b); He et al. (2025; 2024) is proving transformative across multiple AI domains. In video generation, these approaches are crucial for synthesizing dynamic scenes with enhanced spatial and temporal consistency Miao et al. (2025); Yu et al. (2024); Lin et al. (2025); Jiang et al. (2025); Zhang et al. (2024a). World models benefit significantly from 3D/4D representations Zhen et al. (2025); Team et al. (2025), enabling a deeper understanding and prediction of environmental dynamics and underlying physics. For embodied AI, 3D and 4D awareness is fundamental Zhu et al. (2024b;a), improving agent capabilities in navigation Szot et al. (2021) and manipulation Zhu et al. (2024b;a); Lu et al. (2025); Xue et al. (2025); Ze et al. (2024); Fang et al. (2023); Wang et al. (2024a); Yang et al. (2025); Jia et al. (2024). To the best of our knowledge, DeepVerse is the first to incorporate 4D representations into auto-regressive world models.

## 5 CONCLUSION

In this paper, we present DeepVerse, the first interactive world model based on 4D autoregressive video generation. We innovatively introduce 4D representation as our temporal observation to approximate the world's environment. Our experimental results demonstrate the architectural effectiveness of the proposed model and quantitatively confirm the enhancements in visual quality and spatial capabilities achieved through the novel integration of 4D representation. Building upon this, we are capable of achieving long-duration inference and sustaining long-term memory capabilities.

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

# A TRAINING DETAILS

## A.1 FINAL ARCHITECTURE

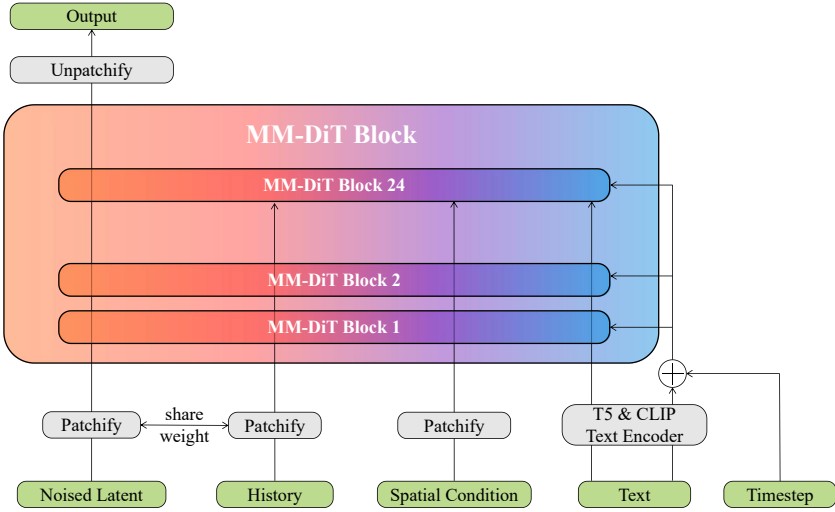

Figure 8: Final architecture.

Our final architecture employs a token-wise concatenation method. As illustrated in figure 8, the noised latent and recent history undergo identical patchify operations to be encoded into tokens, which are subsequently concatenated along the token sequence dimension. Additionally, the spatial condition is independently processed through a patchify operation for token encoding. In alignment with SD3 Esser et al. (2024), our text encoder integrates both T5 Raffel et al. (2023) and CLIP Radford et al. (2021) frameworks, with the obtained embeddings and pooled embeddings being injected into the model through token-wise concatenation and AdaLN mechanisms respectively. To ensure training stability, the MM-DiT Esser et al. (2024) architecture incorporates RMSNorm Zhang & Sennrich (2019) for QK Normalization Henry et al. (2020). The final model outputs are transformed via a linear projection layer to reconstruct tensors matching the shape of the original noise latent. We present the parameters of our model in table 3.

Table 3: Architecture parameters.

| | |
|---|---|
| layers | 24 |
| model dimension | 1536 |
| attention heads | 24 |
| head dimension | 64 |
| spatial position embedding | sincos |
| temporal position embedding | RoPE Su et al. (2023) |
| patch size | $2 \times 2$ |

## A.2 RAYMAP

The raymap serves as an over-parameterized encoding mechanism for 3D viewing, generated through a ray-casting process where each pixel in the image plane emits a directional ray originating from the camera's optical center. This representation maintains spatial correspondence with the original image dimensions while containing 6 channels of geometric information: 3 channels encode the ray origin coordinates (equivalent to the camera position in 3D space), and the remaining 3 channels specify the unit direction vectors of each cast ray. Notably, this parametrization preserves sufficient geometric constraints to enable camera parameter recovery through the reconstruction algorithm 2.

## A.3 COMPACT 4D REPRESENTATION

The final 3D VAE architecture adopted in DeepVerse achieves a temporal compression ratio of 8 along the sequence dimension, enabling the prediction of consecutive future observations spanning 8 time steps. While both image and depth modalities are encoded with 16 latent channels through the 3D VAE, the raymap modality resists effective compression

**Algorithm 2:** Raymap to camera parameters conversion.

**Input** : Raymap $c$
**Output** : Camera Parameters $intrinsic, extrinsic$
Estimate camera position $T \leftarrow Ray\_o(Raymap)$
Estimate ray directions $Ray\_d \leftarrow Ray\_d(Raymap)$
Calculate $intrinsic$ from $Ray\_d$
Calculate camera rotation $R$ from $Ray\_d$
$extrinsic \leftarrow R, T$
return $intrinsic, extrinsic$

via this architecture. To address this, raymap data undergoes spatial downsampling through average pooling to match the latent dimensions of the image modality, followed by temporal concatenation. This configuration results in a combined channel count of $80$ ($16 + 16 + 6 \times 8$), with the majority allocated to raymap representation. However, considering the primary learning challenges reside in the image and depth modalities, we implement a keyframe optimization strategy: Only the final observation in each 8-step sequence is retained as the keyframe, with its complete raymap concatenated (6 channels), while intermediate frames' raymaps are generated through linear interpolation of adjacent keyframes. This approach reduces the input dimensionality to 38 channels ($16 + 16 + 6$) while maintaining temporal coherence. The methodological validity stems from two key observations: 1) Construction of globally consistent 4D representations requires only keyframe inclusion rather than full-sequence encoding, and 2) This selective encoding significantly reduces both global memory requirements and model input complexity, particularly beneficial for maintaining computational efficiency in long-term sequence modeling.

## A.4 DATA BATCH

Through data annotation and filtering protocols, we partition all video content into approximately $30{,}000$ non-overlapping video splits, with each split constrained to a maximum of 400 frames. Under our training configuration, we sample $b$ consecutive 57-frame video clips as a single batch. The system pre-specifies the potential quantity of video clips contained within each split. Notably, while video splits maintain non-overlapping boundaries, individual clips within the same split may exhibit temporal overlap. This methodology ultimately yields a curated dataset of $1.5$ million video clips. For enhanced stability during autoregressive training, we implement a GPU partitioning strategy where devices are grouped into clusters of size $8$ – this configuration precisely corresponds to the temporal dimension length of latent representations generated by 3D-VAE processing of 57-frame sequences. Within each group, GPUs are assigned identical input batches but process distinct temporal target segments.

## A.5 TRAINING TARGET

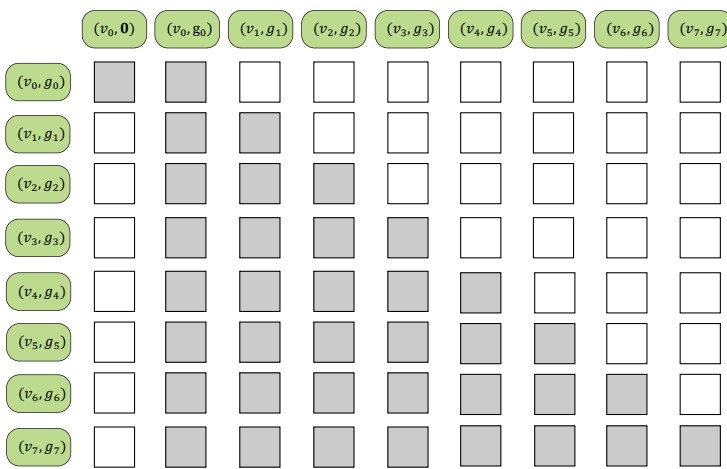

Figure 9: Attention mask during training.

DeepVerse can predict future 4D representations at various subsequent timesteps utilizing only a single input image $v_0$. Since the input image does not constitute a complete 4D representation, we first complete the 4D representation for that timestep. As illustrated in figure 9, we initially predict the complete 4D representation $(v_0, g_0)$ corresponding to the input image, then employ this complete representation to replace the previously incomplete 4D representation. This procedure aligns with and maintains consistency with the inference phase.

**Classifier-Free Guidance Ho & Salimans (2022).** Our framework comprises two distinct condition components: textual condition and spatial condition. During the training phase, we employ stochastic conditioning dropout by masking textual condition $c_T$ with a 10% probability and spatial condition $c_S$ with a 50% probability. For inference, we implement the multimodal conditioning strategy with classifier-free guidance as proposed in InstructPix2Pix Brooks et al. (2023), which coordinates the conditional fusion through learned guidance scales for each modality:

$$
\begin{aligned}
e_\theta(zt, c_T, c_S) = & e_\theta(z_t, ) \\
& + s_T \times \big( e_\theta(z_t, c_T, ) - e_\theta(z_t, , ) \big) \\
& + s_S \times \big( e_\theta(z_t, c_T, c_S) - e_\theta(z_t, c_T, ) \big).
\end{aligned}
\tag{5}
$$

During the inference phase, we employ modality-specific guidance scales of 4 and 5 for textual and spatial condition respectively.

### A.6 TRAINING RESOURCE

To enhance training efficiency, we precomputed and stored text embeddings generated by T5 Raffel et al. (2023) and CLIP Radford et al. (2021) models, thereby eliminating the need to reload these text encoders or reprocess textual inputs during training. The entire training procedure spanned 2 epochs, with our final model requiring approximately $23,000$ A100 GPU hours for completion.

## B EXPERIMENTS DETAILS

### B.1 METRICS

**Fréchet Video Distance (FVD) Unterthiner et al. (2019).** The FVD is a metric used to evaluate the quality of generated videos by measuring the similarity between the distribution of real videos and synthesized videos. It leverages deep features extracted from pre-trained video models to compute the distance between real and generated video distributions in a high-dimensional feature space:

$$
\text{FVD} = \|\mu_r - \mu_g\|^2 + \text{Tr}(\Sigma_r + \Sigma_g - 2(\Sigma_r \Sigma_g)^{1/2})
\tag{6}
$$

where $\mu_r, \mu_g$ are the mean vectors, and $\Sigma_r, \Sigma_g$ are the covariance matrices of real and generated video features. In this paper, we employ I3D networks Carreira & Zisserman (2018) pre-trained on the RGB frame data from the Kinetics-400 dataset Kay et al. (2017) as our feature extraction framework.

**VBench Huang et al. (2024).** VBench serves as a comprehensive evaluation benchmark suite for video generation models, designed to perform systematic assessments. This framework leverages a hierarchical evaluation structure that decomposes the multifaceted concept of "video generation quality" into well-defined constituent dimensions. In this paper, we adopt the six evaluation criteria: *subject consistency*, *background consistency*, *aesthetic quality*, *imaging quality*, *motion smoothness*, and *dynamic degree*, as our primary performance metrics.

### B.2 LONG-DURATION INFERENCE

During the training phase, we have modeled the distribution:

$$
P\big( \hat{s}_{t+1:t+k} \mid a_t, \hat{s}_t, \hat{s}_{t-m:t-1}, \psi\left( \hat{s}_{0:t-m-1} \right) \big).
\tag{7}
$$

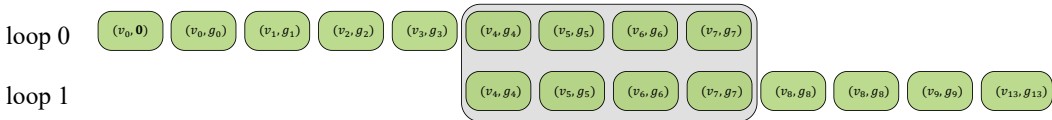

Figure 10: Long-duration inference.

To achieve long-duration inference, when the number of cached observations reaches the predefined *CacheMaxSize* (set as the maximum video clip length during training), we first rescale all cached observations using the preserved $d_{\max}$ parameters. These rescaled observations are then aligned to the global coordinate system through predicted camera parameters and stored in the memory. Subsequently, the most recent $m$ observations are directly adopted as the recent history, as shown in figure 10. The first observation's $d_{\max}$ value within this $m$-length sequence is utilized to scale these $m$ observations, followed by cache updating and subsequent predictions. This methodology ensures global consistency while enabling effective long-duration inference.

## C    LIMITATIONS

Notwithstanding the promising results achieved by DeepVerse, its current framework is trained exclusively on synthetic data, which inherently limits its generalization capability when applied to real-world scenarios. This design choice was made intentionally to isolate the impact of data quality and ensure a controlled experimental setting, thereby allowing a clearer evaluation of the model's intrinsic architecture. However, the domain gap between synthetic training data and real-world testing environments remains a significant constraint. Future work will focus on enhancing the model's robustness and adaptability by incorporating real-world data during training, as well as exploring domain adaptation and generalization techniques to bridge this performance gap.

