# OpenReview forum: "DeepVerse: 4D Autoregressive Video Generation as a World Model"
_ICLR.cc/2026/Conference — ICLR 2026 Conference Withdrawn Submission_

### Official Review · Reviewer_DmCk · 2025-10-21

**Soundness:** 3
**Presentation:** 3
**Contribution:** 3
**Rating:** 6
**Confidence:** 4

**Summary:**

The paper introduces DeepVerse, a 4D autoregressive diffusion model for interactive world modeling. Instead of predicting pixels, it models a unified 4D representation combining RGB, depth, and camera geometry, and maintains a global 4D memory to preserve long-term consistency. The model uses token-wise temporal integration and geometric retrieval to generate coherent long-horizon videos. Experiments show improved spatial consistency and visual quality over PyramidFlow on VBench and FVD metrics.

**Strengths:**

- Large scale model for 4D sequence generation is impressive
- Visual generation results and video consistency improve by adding depth
- The retrieval based on spatial closeness mechanism is simple and effective

**Weaknesses:**

- Missing baselines. No direct comparisons to other works doing 4D modelling. It's easy to show that the model is better than any model not using depth, however it's hard to evaluate author's contribution when there are no comparisons to other models with the same inputs.
- The comparison of the two model architectures (one with channel-wise concatenation, and the other one with token-wise concatenation) is a little confusing. If those two models have different initalization, comparing their final performance is not very fair, which renders this comparison useless.
- The choice of using text as conditioning modality for camera controls isn't ablated. Even though it is convenient, we do not know if it's superior to explicitly conditioning on camera change values.

**Questions:**

- How is textual control mapped to action dynamics in practice?
- Can the memory module handle long-term trajectories beyond short clips?

---

### Official Review · Reviewer_uRwV · 2025-10-31

**Soundness:** 3
**Presentation:** 2
**Contribution:** 3
**Rating:** 2
**Confidence:** 5

**Summary:**

This paper introduces DeepVerse, an autoregressive world model designed to generate coherent long-duration video sequences. The model's key feature is its prediction of a 4D state at each timestep, which includes not only the RGB frame but also a depth map and the 3D camera position. The authors leverage this explicit 3D camera pose to implement a spatial memory mechanism. This mechanism retrieves geometrically relevant past states from a growing memory bank, which are then used as an additional "spatial condition" to guide future predictions. The stated goal of this approach is to enhance long-term spatial and temporal consistency. The model is trained and evaluated on synthetic data derived from game engines.

**Strengths:**

1. The core idea of having a world model generate explicit 3D geometry (depth and camera pose) in addition to pixels is a promising direction for improving long-term consistency in video generation.

2. Using the generated 3D camera pose as a key to retrieve relevant past states from a memory bank is an intuitive and logical approach to help the model "remember" previously visited locations and maintain spatial coherence.

3. The work serves as a reasonable proof-of-concept, demonstrating the feasibility of this 4D-state-based retrieval and conditioning, at least within the confines of synthetic data.

**Weaknesses:**

1. Clarity and Writing: The paper's quality of writing is a significant barrier. The text is often not coherent and difficult to parse, which required substantial and undue effort from the reviewer to understand the proposed methodology and experimental setup.

2. Lack of content: The technical contribution feels limited. A large portion of the paper (e.g., Section 3.1) is dedicated to exploring architectures for conditioning on historical information (channel-wise vs. token-wise). This topic has already been extensively studied in prior work on video generation and diffusion transformers, and its inclusion here feels incremental rather than novel. This space could have been better used to elaborate on the more unique aspects of the method, such as the memory training.

3. Data Dependency and Practicality: A major weakness is the method's reliance on high-quality, metric ground-truth data for depth and 3D camera poses. This data is available in synthetic game environments but is extremely difficult and often impossible to obtain at scale for real-world videos. Real-world depth/pose estimators typically suffer from scale ambiguity and significant noise, which would likely break the proposed geometric retrieval mechanism. This reliance on synthetic data severely limits the practical applicability of the work.

4. Outdated Baselines: The experimental comparisons are not convincing as they rely on outdated baselines. The main comparison is against Pyramid-Flow (2024). In the fast-moving field of video generation, this is no longer a strong baseline. The paper fails to compare against numerous, more recent SOTA models (e.g., Wunyuan, LTX, WAN, etc.). The authors even cite Cosmos in their introduction but do not include it in their experimental comparisons.

5. Qualitative and Quantitative Results: The weak baselines are reflected in the results. The qualitative results (from the supplementary material) appear blurry and are not competitive with current SOTA video generation quality. The quantitative results are also unconvincing. In Table 2, the proposed method does not consistently outperform the Pyramid-Flow baseline, and in several cases, the difference is negligible or even worse. The FVD improvement from the spatial memory shown in Figure 7 is also marginal and the experimental setup for this table (e.g., number of test videos) is not adequately described.

6. Experimental Design: The method's primary theoretical advantage should be in handling trajectories that revisit previous locations. However, the authors explicitly state in Line 407 that "cyclic trajectories are not enforced." This seems like a major missed opportunity to design an experiment that would clearly demonstrate the value of the spatial memory component.

7. Missing Related Work Comparison: The paper fails to discuss or compare with the "GEM: A Generalizable Ego-Vision Multimodal World Model..." (CVPR 2025) paper. While GEM uses camera pose as a control input rather than generating it, it also explores multimodal (RGB+depth) prediction and is thus highly relevant. A discussion of the differences and trade-offs would be necessary.

**Questions:**

1. Could the authors please provide a more detailed explanation of how the spatial condition from memory is used during training? Line 366-368 ("During training, we strategically incorporate this spatial condition at controlled intervals...") is very vague. What percentage of training steps use the memory condition? What does "at controlled intervals" mean?

2. Given the method's strong reliance on metric depth and pose, do the authors have a concrete proposal for scaling this approach to real-world videos? How robust is the retrieval mechanism expected to be against the scale ambiguity and high noise levels produced by current self-supervised pose/depth estimators?

3. What was the reason for not specifically evaluating on cyclic trajectories? This seems to be the ideal test case for the proposed spatial memory. What are the quantitative results (e.g., FVD, consistency metrics) when the model is evaluated on such trajectories, compared to the baseline which lacks this memory?

4. For the quantitative ablation in Figure 7, could the authors please specify the experimental details? On how many videos were these FVD scores computed, and what was the nature of this test data?

---

### Official Review · Reviewer_BZEx · 2025-11-05

**Soundness:** 3
**Presentation:** 3
**Contribution:** 3
**Rating:** 4
**Confidence:** 3

**Summary:**

This paper proposes DeepVerse, a 4D interactive world model that integrates explicit geometric information—such as depth and camera rays—into action-conditioned video prediction. The model combines geometry-aware latent representations with a spatial memory retrieval mechanism to reduce drift and maintain long-term consistency. Through these designs, DeepVerse achieves more stable, realistic, and geometrically coherent long-horizon predictions compared to conventional video-based world models.

**Strengths:**

1. This paper addresses a novel problem: 4D generation and long-term memory, a current hot topic, and the methods, 4D representations (depth + raymap), token-level historical fusion, geometry-based memory retrieval, presented in this work are comprehensive.

2. The paper provides detailed ablation studies on the effect of geometric inputs, spatial conditioning, and historical fusion strategies, supported by both quantitative metrics and visual comparisons.

3. The paper is very well written and easy to read, with a clear logical flow.

**Weaknesses:**

1. The core idea of incorporating geometry and memory into world models has been explored in several concurrent works (e.g., Aether[1], TesserAct[2], WorldMem[3]). The novelty mainly lies in combining these components rather than introducing a fundamentally new mechanism.

2. The paper lacks direct quantitative comparisons with strong recent baselines such as Aether[1], DFoT[4], or FramePack[5].

3.  The geometry-based retrieval module relies on handcrafted distance and orientation metrics, which may be less expressive than learnable attention-based memory mechanisms, WorldMem[3].

[1] Aether Team, Haoyi Zhu, Yifan Wang, Jianjun Zhou, Wenzheng Chang, Yang Zhou, Zizun Li, Junyi Chen, Chunhua Shen, Jiangmiao Pang, et al. Aether: Geometric-aware unified world modeling. arXiv preprint arXiv:2503.18945, 2025.
[2] Haoyu Zhen, Qiao Sun, Hongxin Zhang, Junyan Li, Siyuan Zhou, Yilun Du, and Chuang Gan. Tesseract: Learning 4d embodied world models. arXiv preprint arXiv:2504.20995, 2025.
[3] Zeqi Xiao, Yushi Lan, Yifan Zhou, Wenqi Ouyang, Shuai Yang, Yanhong Zeng, and Xingang Pan. Worldmem: Long-term consistent world simulation with memory. arXiv preprint arXiv:2504.12369, 2025.
[4] Kiwhan Song, Boyuan Chen, Max Simchowitz, Yilun Du, Russ Tedrake, and Vincent Sitzmann. History-guided video diffusion. arXiv preprint arXiv:2502.06764, 2025a.
[5] Lvmin Zhang and Maneesh Agrawala. Packing input frame contexts in next-frame prediction models for video generation. arXiv preprint arXiv:2504.12626, 2025.

**Questions:**

1. Could you provide more comprehensive runtime and efficiency analyses? For example, CUDA time, memory, inference latency, and so on?

2. Since all experiments are conducted on synthetic data, how well does DeepVerse generalize to real-world videos? Have the authors tried fine-tuning DeepVerse on real-world data?

---

### Official Review · Reviewer_paa8 · 2025-11-08

**Soundness:** 2
**Presentation:** 2
**Contribution:** 2
**Rating:** 2
**Confidence:** 5

**Summary:**

This paper proposes DeepVerse—an autoregressive framework for 4D world modeling.

DeepVerse explicitly models temporal and 3D spatial information during the generation process.

DeepVerse directly incorporates 4D information into the generation loop, significantly improving visual consistency in long-sequence generation.

Overall, DeepVerse is both a methodological framework and an interactive world model.

**Strengths:**

This paper integrates 4D explicit modeling into an autoregressive world model and couples the "spatial memory" mechanism with parallel spatial distribution modeling.

The final results of this paper are promising; 4D explicit modeling improves temporal and spatial consistency.

The paper has a clear structure (not related work).

**Weaknesses:**

Confusing and unsupported arguments: Line068 in main paper, "However, these visual-centric strategies fundamentally overlook a critical aspect: videos inherently represent 2D projections of a dynamic 3D/4D physical world. Without explicit modeling of underlying geometric structures, models inevitably struggle to maintain long-term accuracy and consistency in visual predictions," this is the core motivation of the paper; however, such motivation does not cite any references and lacks experimental support. Why is it that without explicit modeling of the underlying geometric structure, the model cannot maintain long-term accuracy and consistency in visual predictions? The video released by Sora-1 in early 2024, "A Woman Walking on the Streets of Tokyo," has been validated by many through 3D reconstruction, demonstrating that the video possesses very good structural information. However, Sora-1 does not perform explicit modeling of the underlying geometric structure, which raises significant issues with the motivation of this paper. The authors cannot prove that the absence of explicit modeling will necessarily lead to failure. I believe this motivation has fundamental flaws.

The second concerning issue is that the title is "4D Autoregressive Video Generation as a World Model." While reading the main body of the article, I assumed that the model architecture was AR-based, constructed by stacking self-attention blocks. It wasn't until I reached Appendix A.1 FINAL ARCHITECTURE that I discovered it is based on a DiT diffusion framework. The term "autoregressive" only applies to how temporal information is processed. In reality, almost all methods handle temporal information in an autoregressive manner. I do not understand why the authors chose to emphasize "autoregressive" in the title while neglecting diffusion.

In my view, the MODEL COMPONENTS lack any novelty. The 4D Representation section simply retrains a VAE on new data; how can this be considered an innovation? Additionally, in the General Control section, you state, 'Our DeepVerse framework deliberately departs from this practice by relying solely on textual conditions.' I understand that you have done this and that it may have some benefits, but for this approach to work, it must be based on certain assumptions.

For example, the diffusion model assumes that data distribution can be modeled through a Markov process of 'adding noise step by step → removing noise step by step,' learning conditional noise or reverse diffusion dynamics to reconstruct the data distribution. The large language model (LLM) assumes that language can be approximated as a massive conditional probability distribution, with a causal direction of 'left to right' or 'masked to visible,' learning useful world and pragmatic structures through maximum likelihood under large-scale parameterization and data, enabling strong generalization.

What are your assumptions? Why do you believe this approach will work? Can you provide a theoretical explanation? I hope you can justify the rationale behind this approach and explain why it is expected to work.

Why is the related work section placed after the experimental section instead of following the introduction? I find this writing style quite strange.

**Questions:**

Why does the spatial position embedding use sinusoidal position encoding while the temporal position embedding uses RoPE? Have you considered using 2D RoPE for spatial position embedding? Or combining spatial and temporal position embeddings together using the 3D RoPE from Cosmos?

You claim that the memories are efficient, but I did not find any evaluation of the model's speed throughout the article. I am unsure how effective the authors consider this model to be in terms of lightweight implementation. If it is very slow, then even the best 4D modeling may have limited practical value.

---

### Note · Authors · 2025-11-14

I have read and agree with the venue's withdrawal policy on behalf of myself and my co-authors.